# Multi-Objective Electric Vehicle Route and Charging Planning with Contraction Hierarchies

**Primary Keywords:** *(1) Applications;*

## Abstract

Electric vehicle (EV) travel planning is a complex task that involves planning the routes and the charging sessions for EVs while optimizing travel duration and cost. We show the applicability of the multi-objective EV travel planning algorithm with practically usable solution times on country-sized road graphs with a large number of charging stations and a realistic EV model. The approach is based on multi-objective A* search enhanced by Contraction hierarchies, optimal dimensionality reduction, and sub-optimal $\epsilon$-relaxation techniques. We performed an extensive empirical evaluation on 182 000 problem instances showing the impact of various algorithm settings on real-world map of Bavaria and Germany with more than 12 000 charging stations. The results show the proposed approach is the first one capable of performing such a genuine multi-objective optimization on realistically large country-scale problem instances that can achieve practically usable planning times in order of seconds with only a minor loss of solution quality. The achieved speed-up varies from $\sim 11\times$ for optimal solution to more than $250\times$ for sub-optimal solution compared to vanilla multi-objective A*.

## Introduction

Multi-objective electric vehicle (EV) route planning addresses the rising problem of long-range trip planning greatly exceeding the vehicle range. Many achievements have been recently presented in this field; however, existing algorithms do not fully address realistic concerns, such as the trade-off between cost and time, large-scale road map, or a large amount of diverse charging stations altogether.

State of the art approaches and algorithms mostly rely on *single-objective* optimization (e.g., Baum et al. 2019a) and are therefore technically limited to always considering only a single objective when finding optimal EV travel plan. Well-established approaches to multi-objective optimization, such as meta-heuristics, can find the Pareto-set only on very small city-sized road networks. Consequently, these approaches are *not* suitable in practice (e.g., Ben Abbes, Rejeb, and Baati 2022). Very recent work of (Schoenberg and Dressler 2023) achieved good planning times while considering multiple simpler objectives (not including cost) on country-scale road networks, but it prohibits planning with a realistic number of charging stations.

Multi-objective EV travel planning is a complex problem (NP-hard but not even in NP) for two main reasons that re-quire both domain-independent and domain-specific techniques to overcome:

1. *Multi-objective optimization*: The problem involves multiple, inherently conflicting objectives (travel time and cost), which inflates the dimensionality of the search space and extends the solution concept from a single solution (route with charging stops) into a Pareto-set of optimal solutions, each with different trade-off of duration and cost).

2. *Integration of charging planning with route planning*: The EV travel planning problem is actually composed of two sub-problems - planning the route in the road network and choosing where and how long to charge. These two problems are closely interconnected, and therefore, we need to solve them holistically to obtain the best solutions.

The multi-objective EV route planning problem we address in this paper is further complicated but, at the same time, more applicable in practice by our use of realistic battery charging (non-linear function), large road networks (country-sized), and different prices and speeds of charging at different charging stations.

In this paper, we show the applicability of multi-objective EV travel planning algorithm based on A* search enhanced by Contraction hierarchies, optimal dimensionality reduction, and sub-optimal $\epsilon$-relaxation techniques with practically applicable solution times on country-sized road graphs with large number of charging stations and realistic EV model. To demonstrate the practical usefulness of our solution, we have set up a prototype application[1], see Figure 1 for an example solution provided by the application. We performed an extensive empirical evaluation of the proposed algorithms on real-world country-scale data with more than 12 000 charging stations involving 182 000 calculated problem instances requiring more than 250 000 CPU hours.

Importantly, our proposed approach is very versatile and can be adapted to other optimization objectives and more complex scenarios, such as time-dependent travel times and charging prices, and therefore presents a generic approach to

---

[1]http://its.fel.cvut.cz/ev-travel-planner. Note that the application does not use precise travel time data (they are expensive) and is for the purpose of potential capabilities and usage demonstration only.

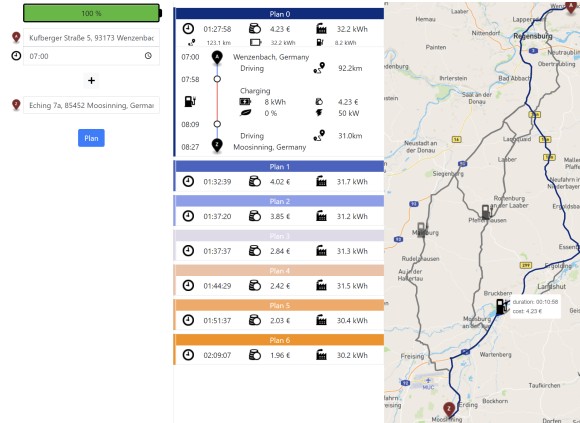

Figure 1: Screenshot of the EV travel planning prototype application. The planning request (left) results in a set of Pareto-optimal plans (middle) and shown on the map (right).

solving a wide range of practical multi-objective EV travel planning problems. We believe our contribution will provide a solid basis for the future exploration of multi-objective approaches to EV travel planning while keeping the practical applicability on large-scale realistic scenarios.

## Related Work

EV travel planning has been first studied with regards to the most energy-efficient routes (Artmeier et al. 2010; Sachenbacher et al. 2011). Schönfelder, Leucker, and Walther (2014) extend the problem of finding the most energy-efficient route by searching not only for a single solution for a given initial state of charge (SoC) but rather for the *consumption profile* function that computes the optimal consumption and route for any possible initial SoC. Storandt and Funke (2012) added to consideration en-route charging, although simplified to always charge to the full battery capacity. Baum et al. (2019b) overcome the need for this simplification by exploiting *consumption profiles* to find the optimal charging options from the virtually infinite number of possibilities due to the continuous nature. All of the above considered only the energy as the optimization objective. Baum et al. (2019a) and Storandt (2012) considered the SoC only as a constraint while optimizing travel time.

Several existing works also extended the problem to multiple objectives. Common approaches for solving multi-objective problems, such as genetic algorithms (Ben Abbes, Rejeb, and Baati 2022) or particle swarm optimization (Siddiqi, Shiraishi, and Sait 2011), were applied to EV travel planning. Although the authors consider the cost of charging, the methods were evaluated only on very small road networks with only hundreds of nodes. Realistic road graphs required in EV route planning have millions of nodes. As such, these techniques do not currently scale to realistic problem instances. Problems on realistically large road graphs are solved by algorithm proposed by Schoenberg and Dressler (2023). The algorithm based on multi-objective A* uses multi-objective adoption of contraction hierarchies (CH)

(Geisberger et al. 2008), that were also used by Baum et al. (2019a). However, they do not consider cost and one of the pre-processing techniques they propose is not suitable for a realistic number of charging stations (12000 in our case). CH were also studied for a bi-objective case by Zhang et al. (2023) but without battery constraints required by EV travel planning. Another potentially applicable method based on pre-processing is presented by Delling and Wagner (2009). The work proposes a multi-objective adaptation of SHARC algorithm (Bauer and Delling 2009) that combines highway hierarchies (Sanders and Schultes 2006) and arc-flags (Möhring et al. 2007) techniques. However, the arc-flag technique is unsuitable for planning with charging stops.

## Multi-Objective EV Travel Planning Problem

We model the EV travel planning problem as a multi-objective constrained shortest path problem with SoC constraints and charging stops with two optimization objectives: time and cost. Formally, we define the EV travel planning problem as a tuple $\mathcal{P} = \langle \mathcal{W}, \mathcal{M}, \mathcal{R} \rangle$ where $\mathcal{W}$ is the global static *EV travel planning environment*, $\mathcal{M}$ is the *EV model*, and $\mathcal{R}$ is the *EV travel planning request* that is specific for each EV user and their needs. The solution to an EV travel planning problem is the Pareto-set of *EV travel plans* $\Pi$.

**EV Travel Planning Environment**   (termed *planning environment* further on) represents the road network and charging stations, i.e., the travel planning components that are independent of the specific details of individual planning requests.

The planning environment is a tuple $\mathcal{W} = \langle G, Q \rangle$, where $G = \langle V, E, \tau, d \rangle$ is a weighted oriented graph representing the underlying road network, with $V$ being the set of nodes representing intersections and $E$ the set of edges representing road segments. Each edge $e = (u, v) \in E$; $u, v \in V$, has a defined traversal duration $\tau(e) \in \mathbb{R}^+$ and a length $d(e) \in \mathbb{R}^+$. [2]

The set of charging stations $Q$ defines the locations where EVs can be charged. Each charging station $q \in Q$ is defined as a tuple $q = \langle v_q, P_q, \gamma_q \rangle$, where $v_q \in V$ is the node where the charging station is located ($V_Q = \{v_q | q \in Q\}$), $P_q \in \mathbb{R}^+$ is the maximum power the charging station provides (*charging rate*) and $\gamma_q : \mathbb{R}^+ \times \mathbb{R}^+ \to \mathbb{R}_0^+$ is the *charging cost function* that defines how much any charging session at the station $q$ costs based on the duration $t \in \mathbb{R}^+$ of the session and the amount of energy $j \in \mathbb{R}^+$ charged during the session. The *charging cost function* can formalize various types of charging policies, including all of those popular today, such as fixed price per charging session, price per minute of charging, price per kWh of charged energy, or their combination.

**EV Model**   $\mathcal{M} = \langle b_{\mathsf{max}}, \beta, \phi, \psi \rangle$ consists of the *maximum battery capacity* $b_{\mathsf{max}} \in \mathbb{R}^+$ of the EV, *cost per km* of driving $\psi \in \mathbb{R}_0^+$, and two functions defining how the EV consumes the energy stored in its battery and how the battery is recharged.

---

[2]Non-essential properties are omitted in the problem definition (e.g., elevation profile required only for the consumption function)

The *energy consumption function* $\beta : E \times [0, b_{\max}] \to [0, b_{\max}] \cup \{-\infty\}$ defines the SoC after traversing edge $e \in E$ while depending on starting SoC. The energy consumption function can take into account various properties of the edge, such as the length or elevation profile. The consumption can be negative due to recuperation. $-\infty$ means that the starting SoC is too low to traverse the edge.

The *charging function* $\phi : [0, b_{\max}] \times (0, b_{\max}] \times \mathbb{R}^+ \to \mathbb{R}^+$ defines the time needed to complete a charging session specified by the starting SoC $b_{\mathsf{start}} \in [0, b_{\max}]$, the final SoC $b_{\mathsf{end}} \in (0, b_{\max}]$ and the maximum available power $P \in \mathbb{R}^+$.

The *cost per km* of driving $\psi \in \mathbb{R}_0^+$ defines EV wear-and-tear costs per driven distance.

**EV Travel Planning Request**  defines the user's specific request for EV travel. The request is defined as a tuple $\mathcal{R} = \langle v_{\mathsf{init}}, v_{\mathsf{goal}}, b_{\mathsf{init}} \rangle$, where $v_{\mathsf{init}} \in V$ is the origin, $v_{\mathsf{goal}} \in V$ is the destination, and $b_{\mathsf{init}} \in [0, b_{\max}]$ is the initial SoC.

**EV Travel Plan**   is a sequence of interleaving states and actions $\pi = (s_0, a_0, s_1, a_1, \ldots, a_{k-1}, s_k)$.

A *state* $s_i$ fully describes the status of the EV and the value of plan objectives at the i-th step of the plan and action $a_i$ describes the transition between the states $s_i$ and $s_{i+1}$. We define the state $s$ as a tuple $\langle v, t, c, b \rangle$ where $v \in V$ is an EV location node, $t \in \mathbb{R}_0^+$ is the time the state is reached, $c \in \mathbb{R}_0^+$ is the charging and driving cost spent to reach the state, and $b \in [0, b_{\max}]$ is the SoC with which the state is reached (higher value means more energy in the battery).

An EV travel plan consists of two types of *actions*:

- *move(e)* that moves the vehicle across the edge $e = (v, u) \in E$:

$$\langle v, t, c, b \rangle \to \langle u, t + \tau(e), c + \psi d(e), \beta(e, b) \rangle$$

- *charge(q, j)* that charges the vehicle at the charging station $q \in Q$ with energy $j \in \mathbb{R}^+$:

$$\langle v_q, t, c, b \rangle \to \langle v_q, t + t_{q,j}, c + \gamma_q(t_{q,j}, j), b + j \rangle$$

where $t_{q,j} = \phi(b, b + j, P_q)$

In order for the EV travel plan $\pi = (s_0, a_0, s_1, a_1, \ldots, a_{k-1}, s_k)$ to be valid, the state of charge must not drop below zero or get above the maximum battery capacity $b_{\max}$: $0 \le b_i \le b_{\max}, \forall i \in 0, \ldots, k$.

We say that an EV travel plan $\pi$ with $k+1$ states is feasible for a planning request $\mathcal{R} = \langle v_{\mathsf{init}}, v_{\mathsf{goal}}, b_{\mathsf{init}} \rangle$ if it is valid, $v_0 = v_{\mathsf{init}}, b_0 = b_{\mathsf{init}}$ and $v_k = v_{\mathsf{goal}}$. We also define the *plan time* as $t_\pi = t_k$ and the *plan cost* as $c_\pi = c_k$.

An EV travel planning algorithm solving problem $\mathcal{P} = \langle \mathcal{W}, \mathcal{M}, \mathcal{R} \rangle$ should produce EV travel plans feasible plans for planning request $\mathcal{R}$ optimal with regard to two objectives – time and cost. More specifically, the goal of the algorithm is to minimize $t_\pi$ and $c_\pi$.

Since there is more than one optimization objective, a total ordering with regard to $t_\pi$ and $c_\pi$ does not usually exist. However, a partial ordering exists according to weak dominance:

**Definition 1** *Let $\pi, \pi'$ be two valid EV travel plans. We say that $\pi$ weakly dominates $\pi'$ (denoted as $\pi \preceq \pi'$) iff $t_\pi \le t_{\pi'}$ and $c_\pi \le c_{\pi'}$.*

Further, we refer to the *weak dominance* only as the *dominance* for simplicity.

**Solution**  to the multi-objective EV travel planning problem $\mathcal{P}$ is a set of feasible Pareto-optimal (non-dominated) EV travel plans $\Pi$. The travel plans are optimal regarding the travel time $t_\pi$ and the cost $c_\pi$ minimization objectives.

## EV Travel Planning Algorithm with CH

To solve the above-outlined problem, we designed an algorithm based on multi-objective A* (Mandow, De la Cruz et al. 2005) enhanced by well-known pre-processing technique Contraction hierarchies (CH) (Geisberger et al. 2008) that reduces the complexity of the route planning part of the problem similarly to Baum et al. (2019a). To further improve the query planning time, we also used optimal *dimensionality reduction* technique (Pulido, Mandow, and Pérez-de-la Cruz 2015) and sub-optimal $\epsilon$-*relaxation* technique (Batista et al. 2011).

Contraction hierarchies that speed up the route planning part of the problem work in two phases. The pre-processing phase assigns a level $lvl(v)$ to each node $v \in V$ and calculates shortcuts $E^{\mathsf{CH}}$ that speed up the query phase. The query phase then performs a search on the graph enhanced with the shortcuts $G^{\mathsf{CH}} = \langle V, E \cup E^{\mathsf{CH}} \rangle$ limited only to *up-down* paths. An up-down path is a path where the level of the nodes is non-decreasing at the first part of the path and decreasing at the rest of the path. If a Pareto-optimal path exists between any pair of two nodes in the original graph $G$, it is guaranteed that an up-down path with the exact same costs also exists in the contracted graph $G^{\mathsf{CH}}$ (Geisberger et al. (2012) reformulated for multiple objectives).

### Pre-Processing Phase

In the pre-processing phase, the nodes of graph $G$ are contracted one by one. When a node $v \in V$ is contracted, for each pair of incoming edge $(u, v) \in E$ and outgoing edge $(v, w) \in E$, a shortcut $e' = (u, w)$ is calculated by their concatenation. The contracted node and its adjacent edges are then removed from the graph. Afterwards, for each shortcut, a *witness search* is started. A *witness search* determines if there exists a *witness* path that dominates the shortcut. If a *witness* path exists, the shortcut is not needed and, therefore, discarded since there exists a better/dominating path. To improve the performance, we calculate the witness search at once for all shortcuts starting at the same node $u$ by a version of multi-objective Dijkstra's algorithm very similar to the algorithm described in Algorithm 1 without heuristics. We also use *hop limit* that bounds the search only to the vicinity of the origin (in our case, to paths consisting of 20 edges at maximum). Although it leads to the addition of unnecessary edges, it does not violate optimality.

The next vertex to contract is determined based on a priority composed of three node metrics proposed by Geisberger et al. (2012) - Edge Difference (ED), Cost of Queries (CQ) and Deleted Neighbors (DN). The resulting priority is $64\mathsf{ED} + \mathsf{CQ} + \mathsf{DN}$ as used by Baum et al. (2019a). The priority is calculated once for all nodes at the beginning of the pre-processing and stored in a priority queue. Furthermore,

we implemented a lazy update of the priority. When a node with minimal priority is polled from the queue, its priority is recalculated, and if the priority is higher than the second smallest priority, the node is reinserted into the queue. The process is repeated until the priority of a node remains the smallest after its update. Additionally, we update the priority of all neighbors of a contracted node. This can easily be done in parallel since they do not change anything until the queue is updated, which can be done in a serial manner after all priorities are calculated. The resulting *contraction order* defines the level of the contracted nodes.

It is not required for all nodes to be contracted. It is commonly used in more complex scenarios (e.g., Baum et al. 2019a) to, for example, lower the number of created shortcuts that, if there are too many of them, could negatively impact the query performance. In our case, we also need it to allow travelling between charging stations required by the need for charging. The set of uncontracted nodes $V^\circ \subset V$ is called the *core* and contains at least all nodes with charging stations $V_Q \subseteq V^\circ$. All nodes in the core have assigned equal level $\forall v \in V^\circ : lvl(v) = |V| - |V^\circ| + 1$.

An edge $(u, v)$ is an *upward* edge iff $lvl(u) \leq lvl(v)$ and *downward* edge iff $lvl(u) > lvl(v)$. An *upward* graph $G^\uparrow = \langle V, E^\uparrow \rangle$ is a graph where all edges $E^\uparrow \subset E \cup E^{\mathsf{CH}}$ are upward while *downward* graph $G^\downarrow$ contains only downward edges $E^\downarrow$.

The edges $e \in E$ in the original graph $G$ of the problem $\mathcal{P}$ have defined three properties - traversal duration $\tau(e)$, distance $d(e)$, and energy consumption function $\beta(e, b)$ which is part of EV model $\mathcal{M}$. Duration and distance of a shortcut created by a concatenation of two edges are trivial. The concatenation of two consumption functions are done by using *consumption/SoC profile* first introduced by Schönfelder, Leucker, and Walther (2014) and used by (Baum et al. 2019a). The SoC profile is a special case of *consumption function* and can be represented by only three values per edge (more details in Baum et al. 2019a). It has also defined dominance relation and therefore allows to easily check dominance of shortcuts and found paths by, e.g., witness search.

## Query Phase

CH queries are commonly solved by bidirectional search algorithms. However, our problem is too complex for easy adoption of bidirectional search, mostly because of the time-dependent nature of charging (dependence on starting SoC) that makes backward search that includes charging very complicated. Therefore, we split the query phase into two sub-phases similarly to (Baum et al. 2019a).

First, we run backward search starting at the destination $v_{\mathsf{goal}}$ on the downward contracted graph $G^\downarrow$ that calculates temporary shortcuts $E^{\mathsf{dest}}$ from the uncontracted *core* (that contains all charging stations) to the destination. This search is based on multi-objective Dijkstra's algorithm very similar to the algorithm used by the witness search in the preprocessing phase. Since the SoC is unknown at the time of the calculation, the algorithm calculates the *SoC profile* instead of just the consumption values.

The second sub-phase runs the multi-objective A\*-based (Mandow, De la Cruz et al. 2005) algorithm described below (pseudocode in Algorithm 1) on the upward graph $G^\uparrow$ with the temporary shortcuts $E^{\mathsf{dest}}$ from the first sub-phase.

**States and Their Dominance** To describe the query algorithm, we use the same definition of states $s = \langle v, t, c, b \rangle$ as presented in the problem definition[3]. As mentioned above, a state can also be viewed as a simpler representation of a (partial) EV travel plan since it fully describes all essential attributes that are necessary for the planning algorithm to decide about the subsequent actions. We say that a plan is *partial* if its last location is not the destination.

In this section, we formally extend the concept of EV travel plan *dominance* (Definition 1) to states while maintaining full compatibility. The algorithm requires two versions of the dominance that are used in different algorithm steps. $\pi$-dominance in Definition 2 is a straightforward adjustment of Definition 1 to the context of states leveraging the information provided by the time and cost heuristics $h_t$ and $h_c$ (described below). The algorithm uses $\pi$-dominance when it checks the explored states against the already found solution plans.

**Definition 2** *Let* $s = \langle v, t, c, b \rangle, s' = \langle v', t', c', b' \rangle$ *be two states. We say that* $s$ $\pi$*-dominates* $s'$ *(denoted as* $s \preceq_\pi s'$*) iff the following conditions are satisfied:*

$$
\begin{aligned}
t &\leq t' + h_t(s') \\
c &\leq c' + h_c(s')
\end{aligned}
$$

However, $\pi$-dominance does not work if both states represent partial plans (not at the destination yet). For example, a state representing a partial plan that is slower but has a higher SoC could lead to a faster plan at the destination because it could have enough energy to reach the destination without any additional stop at a charging station. Therefore, the algorithm requires the following dominance extended by the SoC attribute and without the heuristic estimates to check the states representing partial plans (details how it is used in the section below).

**Definition 3** *Let* $s = \langle v, t, c, b \rangle, s' = \langle v', t', c', b' \rangle$ *be two states at the same node (*$v = v'$*). We say that* $s$ *dominates* $s'$ *(denoted as* $s \preceq s'$*) iff all the following conditions are satisfied:*

$$
\begin{aligned}
t &\leq t' \\
c &\leq c' \\
b &\geq b'
\end{aligned}
$$

At last, we introduce the dominance between a state and a set of non-dominated states.

**Definition 4** *Let* $s$ *be a state and* $S$ *be a set of mutually non-dominated states according to the dominance relation* $\preceq$*. We say that* $S$ *dominates* $s$ *(denoted as* $S \preceq s$*) iff*

$$
\exists s' \in S : s' \preceq s
$$

---

[3]Although the reconstruction of the final plans requires additional state attributes (e.g., a reference to the preceding state and charging details), we omitted them for a clearer presentation.

**Algorithm 1:** Pseudocode of the query phase of the multi-objective EV travel planning algorithm.

---

**Input:** planning environment $\mathcal{W} = \langle\langle V, E^{\uparrow} \cup E^{\mathsf{dest}}\rangle, Q\rangle$
    EV model $\mathcal{M} = \langle b_{\mathsf{max}}, \beta, \phi, \psi\rangle$
    planning request $\mathcal{R} = \langle v_{\mathsf{init}}, v_{\mathsf{goal}}, b_{\mathsf{init}}\rangle$
**Output:** set of Pareto-optimal travel plans $\Pi$

1  $S_v^{\mathsf{op}}$: set of opened states for each node $v \in V$
2  $S_v^{\mathsf{cl}}$: set of visited/closed states for each node $v \in V$
3  $S^{\mathsf{op}} = \bigcup_{v \in V} S_v^{\mathsf{op}}$: set of all opened states
4  $\Pi$: set of solution states
5  $S_v^{\mathsf{op}} \leftarrow \emptyset, \forall v \in V$
6  $S_v^{\mathsf{cl}} \leftarrow \emptyset, \forall v \in V$
7  $\Pi \leftarrow \emptyset$
8  $S_{v_{\mathsf{init}}}^{\mathsf{op}} \leftarrow \{\langle v_{\mathsf{init}}, 0, 0, b_{\mathsf{init}}\rangle\}$
9  **while** $S^{\mathsf{op}} \neq \emptyset$ **do**
10  $\quad$ $s_{\min} \leftarrow \texttt{extractMin}(S^{\mathsf{op}})$
11  $\quad$ **if** $\Pi \preceq_\pi s_{\min}$ **then continue**;
12  $\quad$ **if** $\texttt{inDestination}(s_{\min})$ **then**
13  $\quad\quad$ $\Pi \leftarrow \Pi \cup \{s_{\min}\}$
14  $\quad$ **else**
15  $\quad\quad$ $S_{v_{\min}}^{\mathsf{cl}} \leftarrow S_{v_{\min}}^{\mathsf{cl}} \cup \{s_{\min}\}$
16  $\quad\quad$ $S \leftarrow \texttt{expand}(s_{\min})$
17  $\quad\quad$ **forall** $s = \langle v, t, c, b\rangle \in S$ **do**
18  $\quad\quad\quad$ **if** $b < 0$ **then continue** ;
19  $\quad\quad\quad$ **if** $(S_v^{\mathsf{op}} \cup S_v^{\mathsf{cl}}) \preceq s \vee \Pi \preceq_\pi s$ **then**
20  $\quad\quad\quad\quad$ **continue**
21  $\quad\quad\quad$ **else**
22  $\quad\quad\quad\quad$ $S_v^{\mathsf{op}} \leftarrow S_v^{\mathsf{op}} \setminus \{s' \in S_v^{\mathsf{op}} | s' \preceq s\}$
23  $\quad\quad\quad\quad$ $S_v^{\mathsf{op}} \leftarrow S_v^{\mathsf{op}} \cup \{s\}$
24  **return** $\Pi$

---

**Query Algorithm**  As mentioned above, the optimal algorithm is based on a multi-objective version of A* algorithm guiding the search by two consistent heuristics. We designed remaining travel time heuristic $h_t$ and minimum remaining charging and driving cost heuristic $h_c$.

To further reduce planning times without sacrificing optimality, we employed a technique that significantly reduces the computational complexity of the dominance checks, which are the greatest bottleneck of the proposed algorithm. The *dimensionality reduction* technique described below allows to significantly reduce the size of the Pareto-sets maintained during the search.

The pseudocode of the optimal algorithm is given in Algorithm 1. The algorithm uses four basic types of data structures:

• Pareto-set of visited/closed states $S_v^{\mathsf{cl}}$ for each graph node $v \in V$ that contains all states that were already visited and expanded by the algorithm.[4]

• Pareto-set of opened states $S_v^{\mathsf{op}}$ for each graph node $v \in V$ that holds the states that were generated but not yet visited by the algorithm.[4]

• Solution set $\Pi$ with the states representing plans that reached the destination.

---

[4]The algorithm maintains open and closed sets for all nodes to contain only non-dominated states.

• Set of all opened states $S^{\mathsf{op}} = \bigcup_{v \in V} S_v^{\mathsf{op}}$, that can also be viewed as a priority queue for the states to be visited.

In each iteration, a lexicographically minimal state $s_{\min}$ is extracted from the set of all opened states $S^{\mathsf{op}}$ ($\texttt{extractMin}$ on line 10). The states $s = \langle v, t, c, b\rangle$ are sorted first by their estimated time $t + h_t(s)$, then by cost $c + h_c(s)$ and then by SoC $b$.

Each extracted state is first checked for whether it is not $\pi$-dominated by any of the already found solution states (line 11) and whether it is not a solution itself ($\texttt{inDestination}$ on line 12). If neither is true, the state is added to the corresponding visited/closed set $S_v^{\mathsf{cl}}$ (line 15) and expanded ($\texttt{expand}$ on line 16).

Let $s_{\min} = \langle v, t, c, b\rangle$ be the extracted state. The state is then expanded (function $\texttt{expand}$) using the following actions corresponding to the actions described in the EV travel plan definition:

(i) **move** For each outgoing edge $e = (v, u) \in E^{\uparrow} \cup E^{\mathsf{dest}}$, a new state

$$s = \langle u, t + \tau(e), c + \psi d(e), \beta(e, b)\rangle$$

is generated.

(ii) **charge** For each charging station $q = \langle v_q, P_q, \gamma_q\rangle$ such that $v_q = v$ and for each amount of energy $j$ from a predefined set of target charging levels (for example, charging to 80%, 90%, 100% of battery capacity) a new state

$$s = \langle v_q, t + t_q, c + \gamma_q(t_q, j), b + j\rangle$$

where $t_q = \phi(b, b + j, P_q)$ is generated. The predefined set of target charging levels can be configured arbitrarily, but it should take into account the shape of the charging function $\phi$. For example, if the function is piecewise linear, it should include the breakpoints.

We use the discretization of the target charging levels to significantly reduce the number of newly generated states.[5]

All the newly generated states are first checked if they violate the SoC constraint. If they do, they are pruned immediately (line 18). Then, they are checked if they are not dominated by any of the states in their corresponding $S_v^{\mathsf{op}}$ and $S_v^{\mathsf{cl}}$ Pareto-sets (line 19). Additionally, they are also checked if they are not $\pi$-dominated by any of the already found solution states in $\Pi$. If they are not dominated, they are added to the opened set $S_v^{\mathsf{op}}$ while removing all states in the opened set dominated by the newly generated one (lines 22-23).

**Remaining Travel Time Heuristic**  This heuristic relaxes the battery constraints and estimates the minimum time needed to reach the destination regardless of the battery constraints. It calculates a lower bound on the travel time to the destination.

---

[5]In theory, the optimal solution of the EV travel planning problem would require the ability to consider any arbitrary target charging level. In practice, however, the user can only choose from a discrete set of target charging levels when charging the vehicle and the discretization of the target charging level can be considered as part of the definition of the EV travel planning problem. For this reason, and to simplify the exposition, we refer to EV travel planning as optimal as long as it is optimal with regards also to the set of predefined charging levels.

Let $s = \langle v, t, c, b \rangle$ be a state, then the heuristic can be expressed as $h_t(s) = t(v, v_{\text{goal}})$, where $t(v, v_{\text{goal}})$ is the minimum travel time needed to drive from $v$ to $v_{\text{goal}}$.

We pre-calculate the heuristic using a backward single-objective Dijkstra's algorithm.

**Minimum Remaining Charging and Driving Cost Heuristic** Since the cost objective comprises two components - the charging cost and the driving cost - the heuristic is based on the combination of the lower bounds of both individual components. The calculation of the minimum cost spent on charging is based on the most energy-efficient route to the destination, while the minimum driving cost is based on the length of the shortest route.

Let $s = \langle v, t, c, b \rangle$ be a state, then the heuristic can be expressed as $h_c(s) = b_{\text{min}} c_{\text{min}} + \psi d(v, v_{\text{goal}})$, where $b_{\text{min}}$ is the minimum amount of energy that has to be charged to reach the destination (details below), $c_{\text{min}}$ is the minimum possible price per amount of energy achievable with regards to the cost functions of all charging stations and the charging function of the EV, and where $d(v, v_{\text{goal}})$ is the length of the shortest path from $v$ to $v_{\text{goal}}$. The minimum amount of energy that has to be charged to reach the destination $b_{\text{min}} = \beta(v, v_{\text{goal}}) - b$ is the amount of energy required by the most energy efficient route from $v$ to $v_{\text{goal}}$ deducted by the current SoC $b$.

We pre-calculate the heuristic by a backward label-correcting version (due to the negative consumption) of single-objective Dijkstra's algorithm.

**Dimensionality Reduction** The greatest bottleneck of our proposed algorithm is the computational complexity of dominance checks that is directly dependent on the size of the Pareto-sets managed by the algorithm ($S_v^{\text{op}}$, $S_v^{\text{cl}}$, and $\Pi$). The size of the Pareto-sets can grow exponentially with the size of the problem (in particular, with the size of the road graph and the number of charging stations) and the number of components on which the dominance is based, making the dominance checks very expensive.

Fortunately, we can leverage a technique proposed by Pulido, Mandow, and Pérez-de-la Cruz (2015) that reduces the dimension of some of the Pareto-sets without loss of optimality. If we use the lexicographical ordering for the minimal label $s_{\text{min}}$ extraction (line 10 in Algorithm 1) and if the heuristic estimates $h_t$ and $h_c$ are consistent, we can remove the first attribute (in our case the time) from the dominance checks against the solution set $\Pi$ (line 11) and against the closed set $S_v^{\text{cl}}$ (line 19). Unfortunately, it does not apply to the opened set $S_v^{\text{op}}$.

**$\epsilon$-relaxation** Since the optimal version of our EV travel planning algorithm is too slow (see the experiments), we employed $\epsilon$-*dominance relaxation* (Batista et al. 2011) of dominance conditions to achieve practically usable planning times. For example, $\preceq$ dominance from Definition 3 is extended to:

**Definition 5** *Let $s = \langle v, t, c, b \rangle, s' = \langle v', t', c', b' \rangle$ be two states and $\epsilon_t, \epsilon_c, \epsilon_b \in [0, 1]$ be relaxation ratios. We say that $s$ $\epsilon$-dominates $s'$ (denoted as $s \preceq_\epsilon s'$) iff the following conditions are satisfied:*

$$\begin{aligned}
\epsilon_t \cdot t &\leq t' \\
\epsilon_c \cdot c &\leq c' \\
b &\geq \epsilon_b \cdot b'
\end{aligned}$$

All variants of dominance defined above can be adapted in a similar fashion. This technique is compatible with both heuristics and also with dimensionality reduction.

This relaxation speeds up the algorithm by pruning more states during the search; however, it does not maintain optimality. Therefore, the ratios need to be selected carefully to achieve a good trade-off between the reduction of the planning time and the loss of solution quality.

## Experiment Problem Instances

The EV planning environments used for the evaluation were constructed from real-world data sets for Germany. Germany has a large road network with many charging stations and good accessibility of data. Besides the large-scale Germany area, we also performed the evaluation on a smaller-scale area of the German state of Bavaria.

In the road network graphs created for the experiments, we also included so-called *residential* roads (unlike, for example, Schoenberg and Dressler (2023)) that are important only for the first and last miles. However, they significantly increase the size of the graph and, therefore, increase the complexity of the problem.

We extracted road *graphs* for both Germany and Bavaria from OpenStreetMaps[6] and then mapped real-world charging stations[7] to them. The elevation data were gathered from SRTM.[8]

In total, we experimented with four planning environments. Two Germany environments with 12633 charging stations and 4M nodes or 1.5M nodes without residential roads. The Bavaria environments comprises 2225 charging stations and a road graph with 800k nodes (resp. 300k).

Each charging station in the dataset is described by its location (GPS), the maximum power (kW), and the pricing policy. The pricing policies are of three types: energy-based, duration-based, and fixed. The pricing policy of a charging station can also be a combination of multiple types of policies. They also vary a lot between charging stations, implementing the so-called *location-of-use* pricing.

We model the *energy consumption* of the EV with a linear model that takes into account the length and the elevation profile of the roads similarly to Eisner, Funke, and Storandt (2011). We set the model to approx. correspond to 250 km range with 40 kWh *battery capacity*. We used a piecewise linear *charging function* similar to Baum et al. (2019b) that expresses well the decreasing charging speed when the state of charge approaches the maximum battery capacity while maintaining simplicity. The charging speed gradually decreases starting at 80% battery capacity with other breakpoints at 85%, 90%, and 95%. The charging speed is $6.6\times$

---

[6]https://download.geofabrik.de/europe/germany.html
[7]https://chargemap.com
[8]https://www2.jpl.nasa.gov/srtm/

slower while above 95% than below 80%. The *cost per km* is set to 3 cents per km.

We generated the planning requests as 1000 random *origin-destination* pairs, uniformly sampled from road graph nodes, for both non-residential graphs and then mapped the OD pairs to the full graphs to correspond to the same locations. The origin-destination pairs were generated so that the direct distance was at least 250km (Germany avg. 409km, max. 773km; Bavaria avg. 280km, max. 399km). The initial SoC was set to 100% of the battery capacity.

The algorithm is configured to generate new states for charge action to the following target charging levels based mostly on the used charging function breakpoints: $\{10\%, 20\%, 30\%, \dots, 80\%, 85\%, 90\%, 95\%, 100\%\}$.

## Experiment Results

We implemented our EV travel planning algorithms in Java 17. We ran the experiments on the OpenJDK 64-Bit Server VM Temurin-17.0.4 JVM on a computing cluster node with 64 cores/128 threads 3.1GHz (2 x AMD EPYC 7543). We ran multiple instances simultaneously while limiting the resources to 8 threads and 31GB of RAM per query and to 24 threads and 450GB of RAM per CH pre-processing. Due to the high complexity of the problem, we also introduced a time limit for the query phase to 2 hours.

### Pre-Processing Evaluation

First, we evaluate the impact of the CH core size (# uncontracted nodes). We evaluate the query performance on the fastest optimal configuration of the algorithm, i.e. with both heuristic and dimensionality reduction, but without $\epsilon-$relaxation.

In Table 1, we can see that CH speeds-up queries significantly. On the smaller Bavaria area where it is capable to solve nearly all instances, the speed-up is more than $6\times$. On Germany, we need to look first at the number of solved instances. On the non-residential graph, the algorithm without CH solves 82% of the instances (compared to 98% with CH), and on the full graph it solves only 42% of instances. Therefore, direct comparison avg. query time does not have much value. If we compare only the instances that both algorithms can solve, the speed-up on these simpler instances is $\sim 11\times$ on full Germany ($\sim 7\times$ on non-residential Germany). We can assume that more complex instances benefit more from CH.

We can also see that too great or too small core size negatively impacts the performance. The best optimal query performance can be seen around core size of 75k for full Germany and 40k for non-residential Germany (7k and 5k on Bavaria). Although the average query time on full Germany with core size 75k is slightly slower than the rest, it is capable of solving more instances within the time limit. The additional solved instances, which are the more complex ones, probably cause the greater average query time. Besides longer pre-processing time, it appears that too small core also leads to a dramatically increased number of created shortcuts that slow down the query algorithm by exploring too many unnecessary shortcuts.

| | | $|V^\circ|$ | $t_{pr}$ [min] | $|E^{\mathsf{CH}}|$ | # solved | $t_{avg}$ [s] |
|---|---|---|---|---|---|---|
| Bavaria | Non-residential | 3k | 11.3 | 746k | 1000 | 11.1 |
| | | 5k | 2.3 | 651k | 1000 | 9.6 |
| | | 7k | 1.4 | 615k | 1000 | 10.3 |
| | | 10k | 1.0 | 583k | 1000 | 11.1 |
| | | 15k | 0.8 | 547k | 1000 | 13.5 |
| | | 301k | - | 0 | 1000 | 73.9 |
| | Full | 3k | 383.6 | 2.2M | 997 | 136.5 |
| | | 5k | 33.5 | 1.9M | 999 | 113.6 |
| | | 7k | 13.0 | 1.8M | 1000 | 118.8 |
| | | 10k | 7.0 | 1.7M | 999 | 115.9 |
| | | 15k | 4.5 | 1.6M | 999 | 126.9 |
| | | 811k | - | 0 | 964 | 700.9 |
| Germany | Non-residential | 20k | 52.8 | 3.6M | 980 | 570.6 |
| | | 30k | 20.5 | 3.3M | 981 | 542.4 |
| | | 40k | 10.9 | 3.1M | 982 | 549.0 |
| | | 50k | 9.6 | 3.0M | 980 | 554.6 |
| | | 75k | 7.8 | 2.8M | 972 | 565.2 |
| | | 100k | 7.0 | 2.7M | 971 | 622.9 |
| | | 1.5M | - | 0 | 826 | 1105.1 |
| | Full | 20k | 1216.6 | 10.5M | 745 | 1242.6 |
| | | 30k | 224.0 | 9.4M | 772 | 1258.9 |
| | | 40k | 92.2 | 8.9M | 776 | 1202.8 |
| | | 50k | 68.3 | 8.6M | 774 | 1164.4 |
| | | 75k | 46.6 | 8.2M | 782 | 1221.5 |
| | | 100k | 42.6 | 7.9M | 766 | 1261.8 |
| | | 4.1M | - | 0 | 422 | 1976.5 |

Table 1: Experiment results of the pre-processing phase. $|V^\circ|$ - core size, $t_{pr}$ - pre-processing time, $|E^{\mathsf{CH}}|$ - number of created shortcuts, # solved - number of successfully solved queries, $t_{avg}$ - average query planning time

The main reason why the optimal queries are much slower on the more dense full road graphs is the dramatically increased number of Pareto-optimal solution plans. In the case of Germany, the average Pareto-set size increases from 700 to 1400 (300 to 800 on Bavaria).

### Query Evaluation

We also evaluate the impact of the query phase configurations. We have tried various values of $\epsilon$ coefficients. For simplicity, we present here only the configurations where all $\epsilon$ coefficients are set to the same value ($\epsilon_t = \epsilon_c = \epsilon_b$).

For each planning environment, we first tried the CH pre-processing, which appears to have the best performance with the optimal query algorithm. However, the CH best for optimal planning are not always the best with $\epsilon$-relaxation. For example, on full Germany, the best optimal CH with core size 75k has an average query time of 9.4s with $\epsilon = 0.9$, while CH with core size 50k avg. query time is 8.6 (the difference in max. times is greater - 25.1s vs. 20.7s).

Therefore, in Table 2, you can see the results on the CH with the fastest avg. query times with $\epsilon$-relaxation - 40k for Non-residential Germany, 50k for full Germany, 5k for non-residential Bavaria and 7k for full Bavaria.

We can see that, surprisingly, the fastest avg. query times on full Germany are not achieved with the most aggressive relaxation with $\epsilon = 0.9$ but with $\epsilon = 0.93$. The avg. query

| | | $\epsilon$ | $t_{avg}$ [s] | $t_{max}$ [s] | Avg. $|\Pi|$ | Avg. err |
|---|---|---|---|---|---|---|
| Bavaria | Non-residential | - | 9.6 | 406.7 | 294 | 0.00 |
| | | 0.995 | 0.7 | 3.4 | 32 | 0.02 |
| | | 0.990 | 0.5 | 2.2 | 26 | 0.02 |
| | | 0.980 | 0.5 | 1.5 | 21 | 0.04 |
| | | 0.960 | 0.6 | 1.1 | 17 | 0.06 |
| | | 0.930 | 0.4 | 0.7 | 12 | 0.08 |
| | | 0.900 | 0.4 | 0.6 | 10 | 0.10 |
| | Full | - | 118.8 | 5810.1 | 796 | 0.00 |
| | | 0.995 | 2.0 | 9.0 | 45 | 0.02 |
| | | 0.990 | 1.8 | 6.0 | 33 | 0.03 |
| | | 0.980 | 1.7 | 3.8 | 25 | 0.05 |
| | | 0.960 | 1.5 | 2.4 | 19 | 0.08 |
| | | 0.930 | 1.3 | 2.5 | 12 | 0.11 |
| | | 0.900 | 1.3 | 1.8 | 10 | 0.13 |
| Germany | Non-residential | - | 549.0 | 6809.7 | 713 | 0.00 |
| | | 0.995 | 11.7 | 219.5 | 48 | 0.01 |
| | | 0.990 | 7.5 | 120.7 | 37 | 0.02 |
| | | 0.980 | 5.3 | 64.2 | 29 | 0.03 |
| | | 0.960 | 3.9 | 28.8 | 22 | 0.04 |
| | | 0.930 | 3.2 | 13.2 | 15 | 0.06 |
| | | 0.900 | 2.6 | 8.6 | 13 | 0.08 |
| | Full | - | 1164.4 | 6710.4 | 1422 | 0.00 |
| | | 0.995 | 24.1 | 395.2 | 62 | 0.02 |
| | | 0.990 | 16.6 | 231.7 | 46 | 0.02 |
| | | 0.980 | 13.0 | 114.7 | 34 | 0.04 |
| | | 0.960 | 10.2 | 63.3 | 25 | 0.06 |
| | | 0.930 | 8.2 | 26.0 | 16 | 0.08 |
| | | 0.900 | 8.6 | 20.7 | 13 | 0.10 |

Table 2: Experiment results of the query phase. $\epsilon$ - coefficient used for the $\epsilon$-relaxation speed-up, $t_{avg}$ - average query planning time, $t_{max}$ - maximum query planning time, Avg. $|\Pi|$ - average solution Pareto-set size, Avg. err - average quality loss compared to optimal baseline on instances where both algorithms find the solution

time difference is relatively small (8.2s vs. 8.6s), and the standard deviation for $\epsilon = 0.93$ is 2.5, which, compared to other measured datasets where the deviation is below 2, indicates that this is probably caused by a noise in this dataset. On full Germany, we can reach very good avg. planning times (below 10s), and if we exclude the residential roads (that could be dealt with by post-processing), we can get even to avg. planning times of 2.6s while the maximum is below 10s which is very good for real-world applications.

Because the $\epsilon$-relaxation technique does not preserve optimality, we need to measure also its impact on the *solution quality loss*. We measure the *solution quality loss* as the closeness of the resulting set of EV travel plans to the optimal Pareto-set of plans as proposed by Hrnčíř et al. (2016): $d(\Pi^*, \Pi) = \frac{1}{|\Pi^*|} \sum_{\pi^* \in \Pi^*} \min_{\pi \in \Pi} d(\pi^*, \pi)$. The average distance of the Pareto-set $\Pi$ from the full Pareto-set $\Pi^*$ measures the average Euclidean distance in the objective space (in our case time and money cost) normalized to $[0, 1]$ range. For each objective, the minimum value from all plans $\Pi \cup \Pi^*$ is mapped to 0, and correspondingly, the maximum value is mapped to 1. For illustration, if the optimality loss was 7% equally distributed among the objectives, the distance

$d(\pi^*, \pi)$ would be approx. 0.1.

More aggressive pruning leads to smaller solution Pareto-sets (avg. 13 plans with $\epsilon = 0.9$). While the sub-optimal Pareto-sets are dramatically smaller the solution quality loss is very reasonable (below 0.1), which means that the diversity of the Pareto-optimal set of plans is covered well by the subset obtained with $\epsilon$-relaxation. We can also see that the inclusion of the residential roads does not significantly increase the diversity of the solution since the average sizes of the relaxed solutions that maintain good diversity are very similar with or without residential roads.

Figure 2 illustrates the trade-off between the achieved speed-up and the solution quality loss (and also the surprising behavior described above). We can see that on full Germany a speed-up of $\sim 150\times$ (on non-residential even $\sim 170\times$) can be achieved with only a minor solution quality loss of 0.08.

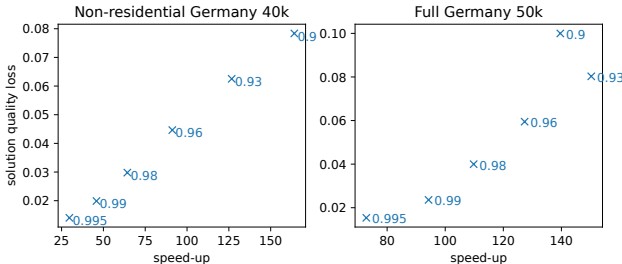

Figure 2: The trade-off between average solution quality loss and average speed-up achieved by $\epsilon$-relaxation. The $\epsilon$ coefficients are displayed directly in the plot.

## Conclusions

In this paper, we show the applicability of multi-objective EV travel planning algorithm based on A* search enhanced by Contraction hierarchies, optimal dimensionality reduction, and sub-optimal *$\epsilon$-relaxation* techniques on realistic country-sized road graphs with a large number of charging stations and realistic EV model.

Our extensive evaluation demonstrated the great impact of CH speeding up the optimal algorithm on the most complex scenarios more than $\sim 11\times$. Together with the sub-optimal relaxation, with which the additional speed-up is $\sim 150\times$, the algorithm solves instances on Germany with residential roads below 9 seconds on average while maintaining a very good solution quality and providing 13 plans on average. On the less dense Germany graph without residential roads, even the maximum time is below 9 seconds. The total combined speed-up is more than $250\times$ (on the 422 instances the A* algorithm without CH can solve). These results prove that the proposed approach is the first one capable of performing such a genuine multi-objective optimization on realistically large country-scale problem instances that can achieve practically usable planning times in order of seconds with only a minor loss of solution quality.

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
