# OpenReview forum: "Multi-Objective Electric Vehicle Route and Charging Planning with Contraction Hierarchies"
_icaps-conference.org/ICAPS/2024/Conference — ICAPS 2024_

### Official Review · Reviewer_XT4N · 2024-01-20

**Significance And Importance:** 2
**Soundness:** 3
**Novelty:** 2
**Clarity:** 4
**Overall Evaluation:** 1
**Confidence:** 5

**Weaknesses:**

1: Minor weaknesses that are easily fixable.

**Contributions Of The Paper:**

The paper considers the problem of electric vehicle route planning, taking charging and realistic aspects as non-linear charging functions into account. The main contribution is to combine several known preprocessing and query acceleration methods to make the overall approach usable and scalable. The experimental evaluation illustrates  the  trade-offs between query time and route quality that can beachieved.

**Ethical Considerations:**

(1) Not Applicable: The paper does not have any ethical considerations to address

**Nomination For Best Paper:**

No

**Questions For Authors:**

Could you please elaborate on W3?

**Reproducibility:**

3: Authors describe the implementation and domains in sufficient detail.

**Strengths Of The Paper:**

S1. The framework is a nice piece of engineering and neatly shows how to integrate different approaches to solve a complex real-world planning problem.

S2. The paper is well written, the problem definition is clean and most algorithms are described in sufficient detail.

S3. The evaluation is comprehensive and was conducted on real-world data.

**Weaknesses Of The Paper:**

W1. The paper is quite incremental. CH is used for multi-criteria route planning, including the consideration of edge cost functions and charging functions in particular, for a long time. The heuristics are not new, and the query algorithm closely follows the approach by Baum.

W2. Despite discretization and relaxation steps, the resulting query times are still quite large for interactive navigation.

W3. It is stated in the problem definition that recuperation may induce negative energy consumption along an edge and in the description of the cost heuristic, this aspect is included. However it is not sufficiently explained how this is dealt with in the actual CH construction. There, it is stated that multi-objective Dijkstra is used to compute witness paths. But with negative costs the label-setting property of the algorithm is harmed. There are methods to restore that property using additional preprocessing as described by Eisner et al (2011), but it is unclear whether this is used here or whether  also a label-correcting Dijkstra variant is applied in that phase.

W4. The demo is not convincing. Of course it is much appreciated that a demo is linked at all and the interface is really nice. But based on testing it, there are some doubts about the validity of the method. Queries are quite slow even if source and target are in short range. For most tested queries the charging cost was zero despite there being one or even multiple charging stops. This leads to some quite cumbersome route options.


Minor comments:
- l. 216 'EV travel plans feasible plans'
- It is a bit strange to emphasize how many queries were done and how many CPU hours that took. More queries does not necessarily mean more expressive results and more CPU hours could also mean a slow approach.

---

> ### Author Rebuttal · Authors · 2024-01-26
>
> Thank you for the review and for pointing out missing details.
>
> Regarding W2:
> For our experimental and easy development purposes, we needed the implementation to be more general, versatile, and easily configurable to perform all the experiments. Therefore, there is a space to improve the performance of the algorithm just by a more specialized implementation that could also be tuned for the specific algorithm configuration. We believe that just with this specialized implementation, we could decrease the execution time at least by 30%. We also believe that there are other possibilities to improve the algorithm itself, e.g., we would like to focus in the future on more informative heuristics.
>
> W3:
> In our case, the CH construction can use label-setting multi-objective Dijktra’s algorithm. Meaning that if a state is polled from the priority queue there cannot appear another state on the same node that dominates it. The algorithm is very similar to Algorithm 1 without heuristics. If a state ‘s’ is polled from the queue with lexicographical ordering (our case), it means that all other states in the queue have either (1) a worse time or (2) have equal time and worse distance or (3) both time and distance are equal, and the consumption is equal or worse.
> From states (1) and (2), states dominating ‘s’ cannot be produced since both time and distance are positive. Consumption can be negative and, therefore, can be improved; however, even states (3) cannot generate states dominating ‘s’ since it is not possible to move without increasing the time and distance; hence, the label-setting property is not harmed.
>
> W4:
> Unfortunately, we cannot run the demo on state-of-the-art HW on which we calculated the experiments. However, we moved it to a slightly better server and made configuration changes in the algorithm running on the backend. Therefore, the performance should be better now. Regarding zero-cost charging, the real-world data we use contains approx. 30% of free charging stations. Therefore, they are commonly used in the displayed cheaper travel plans.
>
> Regarding the amount of queries and CPU hours:
> We think that one of the contributions is the extensive experiments. We empirically proved the algorithm properties, and that could not be reliably done without a great number of queries that also, with the less performant configurations, lead to great requirements of computational resources.

---

### Official Review · Reviewer_F821 · 2024-01-21

**Significance And Importance:** 2
**Soundness:** 3
**Novelty:** 3
**Clarity:** 4
**Overall Evaluation:** 2
**Confidence:** 4

**Weaknesses:**

1: Minor weaknesses that are easily fixable.

**Contributions Of The Paper:**

The paper proposes an enhanced A* search for a multi-objective EV route planning problem.

**Ethical Considerations:**

(1) Not Applicable: The paper does not have any ethical considerations to address

**Nomination For Best Paper:**

No

**Questions For Authors:**

- How easily could more complex problem characteristics (eg, time windows on visits) be incorporated into the method? What problem characteristics would hinder the algorithm?

**Reproducibility:**

4: Authors promise to release code and domains (whichever apply).

**Strengths Of The Paper:**

The paper is well-written, clearly organized, and relevant for an ICAPS audience. The problem is interesting and the algorithm is effective at solving real-world scale problems.

**Weaknesses Of The Paper:**

- It is not immediately clear why the travel time and cost objectives are conflicting. Wouldn't a shorter trip result in less cost? (less charge consumed). The reviewer recommends this be clarified more thoroughly.
- The literature review does not acknowledge the massive amount of work on electric vehicle routing problem (EVRP) [1] in the Operations Research literature, including related techniques for multigraph preprocessing [2] etc.
- Page 3: "EV travel plans feasible plans" seems like a typo.

References:
[1] Schneider, Michael, Andreas Stenger, and Dominik Goeke. "The electric vehicle-routing problem with time windows and recharging stations." Transportation science 48.4 (2014): 500-520.
[2] Froger, Aurélien, et al. "The electric vehicle routing problem with capacitated charging stations." Transportation Science 56.2 (2022): 460-482.

---

> ### Author Rebuttal · Authors · 2024-01-26
>
> Thank you for the review and for pointing out missing details.
>
> Answer to the questions:
> If you mean time windows on charging stations (occupancy), the algorithm would require only minor modifications. During the search, waiting times until the first available time window have to be added. If you mean to add waypoints en route that must be visited within a specified time window, the algorithm would have to “connect” all the waypoints to the contraction hierarchies core similarly as it does with the destination. Additionally, it would have to calculate also the routes from each waypoint to the core and not only the routes from the core to the waypoint. This could be easily done in parallel for each waypoint so it would not significantly increase total computation times. The algorithm can also easily deal with time-dependent charging prices, and there are also versions of contraction hierarchies for time-dependent travel times [3] and frequent updates [4].
>
> Regarding weaknesses:
> If the trip would be short enough to not require charging, this would be commonly true. However, on long-distance trips requiring multiple chargings, the objectives become conflicting since faster charging stations are usually more expensive. Moreover, the fastest routes often use highways that are usually longer than the shortest routes. If the paper is accepted, we would like to add a similar explanation to the introduction.
>
> Generally, VRPs are much more complex than single-source, single-destination route planning problems (our case) and are therefore solved by very different approaches (e.g., metaheuristics as in [1] and [2]). Although EVRPs have to deal with similar concerns (battery, etc.) as we do, unfortunately, the space is limited, and we decided to use it for, in our opinion, more related and closer papers. If the paper is accepted, we could add a few references to this group of problems.
>
> [3] Batz, G. Veit, et al. "Time-dependent contraction hierarchies." 2009 Proceedings of the Eleventh Workshop on Algorithm Engineering and Experiments (ALENEX). Society for Industrial and Applied Mathematics, 2009.
> [4] Geisberger, Robert, et al. "Exact routing in large road networks using contraction hierarchies." Transportation Science 46.3 (2012): 388-404.

---

### Official Review · Reviewer_s1qe · 2024-01-22

**Significance And Importance:** 2
**Soundness:** 3
**Novelty:** 3
**Clarity:** 3
**Overall Evaluation:** 1
**Confidence:** 4

**Weaknesses:**

0: Minor weaknesses requiring some work to be addressed for the paper to be accepted.

**Contributions Of The Paper:**

The paper presents a novel approach to multi-objective EV travel planning, integrating route and charging planning. The approach utilizes a multi-objective A* search enhanced by Contraction Hierarchies for optimal dimensionality reduction and sub-optimal \epsilon-relaxation techniques. This methodology was empirically evaluated on over 182,000 problem instances in real-world settings like Bavaria and Germany, featuring over 12,000 charging stations. The results demonstrate the algorithm's capability to perform genuine multi-objective optimization on large-scale problems, offering a significant speed-up in planning times while maintaining high solution quality. This work contributes to practical, efficient EV travel planning on a country scale.

**Ethical Considerations:**

(1) Not Applicable: The paper does not have any ethical considerations to address

**Nomination For Best Paper:**

No

**Questions For Authors:**

Please answer comments 3 and 4 in the "Weaknesses" box above.
In particular, I would like to know:
1. Whether any evidence of objective conflicts exist, and,
2. How can uncertainties on the actual travel/charging times be handled?

==

I have reviewed the authors' rebuttal and I appreciate their answers. They have addressed most of my earlier questions.

**Reproducibility:**

2: Some details are missing, but the paper still appears to be replicable with some effort.

**Strengths Of The Paper:**

- large real-world-inspired instances.
- efficient and fast solution time.
- good UI front-end that demonstrates how the idea works.

**Weaknesses Of The Paper:**

1. The paper jumps straight into the description of the approach without much motivation and discussion on the design choices. For example, it is not argued clearly why multiple objectives have to be handled by using a multi-objective optimization approach, and not by a single objective function integrating multiple objectives.

2. From the flow of the paper I can see that the authors chose to formulate the problem as presented probably due to their desire to utilize the multi-objective A* algorithm. However, given that the problem can be naturally formulated as an MDP, isn't it more general to first model it as an MDP, and then seek an appropriate solution approach?

3. In the experiment section, the whole focus is on execution time. While this is an important result, I think even more important is the demonstration (quantification) of the actual trade-off between the two objectives. In particular, I think it would be crucial for the authors to clearly demonstrate that the chosen two objectives are indeed significantly conflicting with each other, and warrant the proposed multi-objective approach.

4. The proposed model does not seem to consider real-time updates, or uncertainties resulting in edge travel time and charging time. However, in practice, these congestions are important and could significantly affect the implementation of the computed plan.

5. Minor note: on page 3, line 252, it was mentioned: "If a Pareto-optimal path exists ...". Can there be cases where a Pareto-optimal path does not exist?

---

> ### Author Rebuttal · Authors · 2024-01-26
>
> Thank you for the review and for pointing out missing details.
>
> Regarding weaknesses 1 and 3:
> While driving, time and cost usually correlate; however, the charging is the opposite. Faster charging stations are usually more expensive (avg. in our data - 50kW: 0.7€/kWh, 22kW: 0.5€/kWh), which makes the objectives conflicting (it can be seen in the avg. solution Pareto-set size: 1422 on Full Germany). We agree that it would be beneficial to also show the objective trade-off in the experiments; however, there is only a limited space, and we believe that it is more important to show that it is possible to achieve usable execution times even with the more complex multi-objective approach.
>
> We use the multi-objective approach instead of e.g. weighted sum because the resulting Pareto-set properly explores the trade-off between the objectives instead of a few preselected time-cost weight combinations providing a fixed number of options. Consider trip from Freiburg to Leipzig. The Pareto-set contains 16 options; the fastest plan provided by the demo takes 6h45min and costs 64€ while a plan that takes 1 more minute costs 37€. Another 10€ can be saved by a plan that takes 8 more minutes. Fixed weight combinations could find these options if the weights were tuned to this specific trip, but only a multi-objective algorithm can discover the best trade-offs for all trips.
>
> If the paper is accepted, we would like to add a similar explanation to the introduction.
>
> W2:
> Since similar problems are commonly defined as a search on road graphs and our problem contains no stochasticity, we do not see the benefit of MDP formulation.
>
> W4:
> State-of-the-art navigations work with real-time data. Our algorithm is directly prepared to handle real-time charging station data. The contraction hierarchies pre-processing makes using the real-time traffic data more complicated; however, there are CH modifications that deal with both time-dependent edge costs[1] and frequent updates[2]. We can clarify this in the paper and discuss future work in case of acceptance.
>
> W5:
> It is possible if the road graph is disjoint. It is just a statement that can be reformulated to “For each Pareto-optimal path…”.
>
> [1] Batz, G. Veit, et al. "Time-dependent contraction hierarchies." Proceedings of the Eleventh Workshop on Algorithm Engineering and Experiments (ALENEX), 2009.
> [2] Geisberger, Robert, et al. "Exact routing in large road networks using contraction hierarchies." Transportation Science 46.3 (2012).

---

### Meta-Review · Area_Chair_N6bk · 2024-02-04

**Recommendation:** Accept (Oral)
**Confidence:** 4

**Metareview:**

Positive points:
- new approach integrating the route and charging planning for an electric vehicle (EV)
- enhanced A* search for a multi-objective EV route planning problem
- real-world setting with many solved instances, including large-scale problems
- demo application provided, with runtime reactions reasonably fast
- well-written paper with sufficient details

Important issues sorted in the rebuttal:
- needs for multi-objective setting clarified
- demo application: reasonably fast now, zero-cost charging stations included in the data

Future work:
- handling of uncertainties
- performance of the algorithms for interactive navigation can still be improved

We further ask the authors to reflect on the reviews' comments in the final version of the paper in case of final acceptance.

**Ethical Considerations:**

(1) Not Applicable: The paper does not have any ethical considerations to address